# OpenReview forum: "LightBagel: A Light-weighted, Double Fusion Framework for Unified Multimodal Understanding and Generation"
_ICLR.cc/2026/Conference — ICLR 2026 Conference Withdrawn Submission_

### Official Review · Reviewer_KSuX · 2025-10-27

**Soundness:** 3
**Presentation:** 3
**Contribution:** 2
**Rating:** 4
**Confidence:** 4

**Summary:**

This paper proposes LightBagel, a unified multimodal model that couples a pretrained VLM understanding branch with a pretrained DiT generation branch by inserting zero-initialized multimodal self-attention after each block in both branches, allowing ViT tokens to carry high-level semantics while VAE tokens preserve spatial detail. The understanding branch remains frozen, the reported training budget is modest, and the system is competitive on GenEval, DPG-Bench, GEdit-Bench, and ImgEdit-Bench under the stated constraints. Despite these merits, the novelty appears limited because similar unified modeling strategies have already been extensively explored by the community, so the contribution reads more as a careful engineering integration than a substantive conceptual advance.

**Strengths:**

LightBagel presents a clean and practical architecture that interleaves zero-initialized multimodal attention to couple strong pretrained components while preserving their initial capabilities. The method establishes a clear token-level division of labor, with ViT tokens carrying semantics and VAE tokens capturing spatial detail, which aligns well with image editing. Results are competitive relative to the reported training budget, and the paper presents a coherent token-efficiency narrative. Ablations on fusion depth and timestep shift provide concrete and reproducible design guidance.

**Weaknesses:**

1. **Incremental novelty relative to recent unified models:** LightBagel interleaves zero-initialized multimodal attention after each block to connect a frozen VLM and a DiT, but similar deep-fusion and connector-style unification has already been explored in BAGEL and LMFusion, which also preserve pretrained capabilities while enabling generation. To isolate the technical contribution, the paper should add head-to-head ablations against them with strict parity in parameter counts, sequence lengths, training tokens, and data, and analyze where interleaving at matched depth yields measurable advantages over the generalized causal attention or connector designs used previously.

2. **Evaluation clarity and fairness on GenEval:** The reported GenEval number appears to rely on an LLM-rewriter setting, while several baselines mix rewritten and non-rewritten prompts. Since GenEval is sensitive to prompt phrasing, this can inflate relative gains. The paper should report both rewritten and non-rewritten results for LightBagel, mark every baseline consistently.

3. **Insufficient compute and memory accounting:** The method inserts additional attention modules at every layer, yet the paper does not specify hardware, GPU count, training throughput, effective tokens per second, peak memory, or inference latency. Without a cost profile, readers cannot judge efficiency or deployability.

**Questions:**

1. **Novelty and parity:** Could you briefly clarify the concrete technical delta over BAGEL and LMFusion and, if feasible, include one small controlled comparison under roughly matched settings to illustrate when interleaving at matched depth helps?

2. **GenEval settings and fairness:** Please confirm whether your GenEval score uses the LLM-rewriter and whether baselines are aligned. If possible, add a short two-row table showing LightBagel with and without rewriting to indicate sensitivity.

3. **Compute and memory accounting:** Please share indicative numbers for a representative setup (GPU model/count, training hours) and a single-shot inference latency and peak memory at 512 px to contextualize cost.

**Details Of Ethics Concerns:**

N/A.

---

### Official Review · Reviewer_J38T · 2025-10-27

**Soundness:** 3
**Presentation:** 3
**Contribution:** 2
**Rating:** 4
**Confidence:** 3

**Summary:**

The authors proposed a unified multimodal framework that bridges understanding and generation in a single architecture. They build upon two pre-trained backbones (one for visual-language understanding and one for diffusion-based image generation), and subsequently interconnect them via a layer-wise fusion module to allow semantic features learned by the understanding branch to flow into the generation branch (and vice versa), such that both high-level semantic visual features and low-level image latent features are jointly leveraged.

The authors also collected approximately 45 million samples, which are mostly from publicly available datasets, with about 10% supplemented by a synthetic self-curated dataset, to train the model. Empirical evaluations across standard benchmarks show that their model achieves strong performance on both multimodal reasoning and image generation/editing tasks, demonstrating that this kind of “dual-path, dual-fusion” design can efficiently reuse pre-trained models for unified multimodal tasks.

**Strengths:**

1. Layer-wise Bidirectional Fusion Design: Authors introduce a systematic, layer-wise fusion scheme that allows bidirectional information exchange between a frozen vision–language understanding backbone and a diffusion-based generation backbone. This integration is clearly described, implemented, and empirically shown to yield improvements across multiple multimodal tasks.
2. Effective Reuse of Pre-trained Models: The Authors freeze the understanding branch and insert zero-initialized fusion layers, maintaining stability during training.
This provides a practical contribution to multimodal system engineering, though it depends heavily on the availability of strong pretrained backbones.
3. Empirical Validation Across Multiple Benchmarks: The paper reports gains on well-known datasets such as GenEval, DPG-Bench, and GEditBench-EN, indicating that the proposed integration produces measurable practical benefits.

**Weaknesses:**

1. Lack of Theoretical or Representational Analysis: The proposed architecture is more likely an incremental combination of existing pretrained models and fusion layers without introducing a novel learning mechanism. The core contribution feels more like one engineering work, despite that the engineering integration is neat and the results are okay.
2. Data Efficiency: By inserting only minimal fusion modules and freezing most parameters, the model achieves competitive performance with comparatively limited additional training cost. But the claimed “efficiency” is relative. The framework leverages already-expensive pre-trained components rather than achieving efficiency from first-principles learning design.
3. Lack of Causal Ablation and Attribution Clarity: The paper claims gains from “Double Fusion,” but fails to show which part (bidirectionality, layer count, or dual-level features) truly matters.  For example, in order to determine whether bidirectional information exchange between the understanding and generation branches is essential or merely redundant, controlled experiments can be set to compare the proposed method with unidirectional and no-fusion variants.

**Questions:**

1. I am not sure what 50% depth means in Section 4.4. 0% depth represents that i-th DiT block is conditioned on i-th VLM block, and 100% represents that every DiT block is conditioned on the final VLM block. Does 50% depth means every DiT block is conditioned on the layer of the middle depth VLM block?
2. Since ViT tokens are from Qwen2.5-VL and 3D causal VAE from Wan2.2-TI2V-5B, I cannot find them in the experimental results. While authors claimed the proposed method can keep the strong ability of each existing model, it is better to compare LightBagel with them. If I miss the comparison results, authors can point them out in the rebuttals.
3.  Authors have created a synthetic dataset of 4.5 million samples and used it together with some other public datasets to train the LightBagel model. The experimental results combine the effects of both the dual-fusion architecture and the newly curated dataset, making it difficult to isolate the contribution of each factor.

---

### Official Review · Reviewer_EPax · 2025-10-29

**Soundness:** 3
**Presentation:** 3
**Contribution:** 3
**Rating:** 4
**Confidence:** 3

**Summary:**

This paper introduces LightBagel, a light-weight unified multimodal model designed to achieve competitive performance in both multimodal understanding and image generation/editing with significantly reduced training cost.
The key idea is a double fusion mechanism that interleaves multimodal self-attention blocks between pretrained Vision-Language Models and Diffusion Transformers. This architecture—termed “deep fusion”—preserves the original strengths of each base model (autoregressive reasoning in VLM and high-fidelity rendering in DiT) while enabling rich cross-modal interactions throughout the network.
LightBagel employs: A dual-pathway architecture (VLM + DiT) augmented with zero-initialized multimodal self-attention modules for stable integration. A layer-wise fusion strategy connecting high-level ViT features and low-level VAE representations. A carefully curated dataset of 45M multimodal samples (public + synthetic) covering text-to-image and image-editing tasks. Trained with only ~35B tokens, it matches or surpasses larger models such as OmniGen2, BAGEL, and UniPic, while using 10×–20× fewer tokens.

**Strengths:**

1. The paper is well written, with experiments on multiple benchmarks and thorough ablation studies.
2. LightBagel achieves competitive or superior results to large-scale UMMs while using an order of magnitude fewer training tokens, demonstrating excellent token efficiency.

**Weaknesses:**

1. The paper lacks comparison with two base models Qwen and Wan (quantitative/qualitative) to show the improvements of design.
2. The paper focuses heavily on benchmark scores but lacks detailed investigation into why double fusion improves multimodal alignment or where it might fail.
3. The design seems to only improve generation ability with understanding. The authors are actually adding an understanding module to more accurately generate images.

**Questions:**

I believe that a truly unified framework should not just combine an understanding model and a generative model in the most lightweight way, but should enable mutual enhancement between understanding and generation.
In this framework, it seems that only the generative side is being strengthened — what about the other direction?
I will increase my score if this question is addressed.

---

### Official Review · Reviewer_6ety · 2025-10-29

**Soundness:** 2
**Presentation:** 3
**Contribution:** 2
**Rating:** 2
**Confidence:** 3

**Summary:**

This paper proposes an efficient fusion strategy that utilizes publicly available unified multimodal models trained for both understanding and generation.  The paper uses the MoT architecture to enable layer-by-layer feature interaction between understanding models and generation models. The proposed method, LightBagel, achieves competitive performance in comprehension, generation, and editing.

**Strengths:**

1. Unified Multimodal Models are currently a hot topic of research. The training strategy proposed in the paper can efficiently utilize the understanding and generation capabilities of open-source models and achieve unification.
2. The paper's experimental results are competitive and comparable to the performance of current unified multimodal models.

**Weaknesses:**

1. The architecture proposed by the paper has already been discussed in Bagel, and Bagel's experiments demonstrated the architecture's effectiveness. The paper seemingly just trains open-source understanding and generation models following Bagel's structure.
2. Insufficient experiments: The paper's core module adopts an MoT structure to fuse the understanding and generation models, but it provides no comparison regarding the choice of the MoT structure or against other fusion methods.

**Questions:**

See Weaknesses

---

### Note · Authors · 2025-11-12

I have read and agree with the venue's withdrawal policy on behalf of myself and my co-authors.